# Activity Type Effects Signal Quality in Electrocardiogram Devices

**DOI:** 10.3390/s25165186

**Published:** 2025-08-20

**Authors:** Bryndan Lindsey, Samantha Snyder, Yuanyuan Zhou, Jae Kun Shim, Jin-Oh Hahn, William Evans, Joel Martin

**Affiliations:** 1Johns Hopkins Applied Physics Laboratory (JHU/APL), Laurel, MD 20723, USA; bryndan.lindsey@jhuapl.edu; 2Orthopedics and Sports Medicine, MedStar Health Research Institute, Baltimore, MD 21218, USA; samantha.snyder@medstar.net; 3Department of Mechanical Engineering, University of Maryland, College Park, MD 20742, USA; yzhou114@umd.edu (Y.Z.); jkshim@umd.edu (J.K.S.); jhahn12@umd.edu (J.-O.H.); 4Department of Exercise Science, Elon College, College of Arts & Sciences, Elon University, Elon, NC 27244, USA; bevans19@elon.edu; 5Center for the Advancement of Well-Being, George Mason University, Fairfax, VA 22030, USA; 6Sports Medicine Assessment Research & Testing (SMART) Laboratory, George Mason University, Manassas, VA 20110, USA

**Keywords:** wearable ECG devices, electrocardiogram, ECG signal quality, motion artifact, physiological monitoring

## Abstract

**Highlights:**

**Abstract:**

Electrocardiogram (ECG) devices are commonly used to monitor heart rate (HR) and heart rate variability (HRV), but their signal quality under non-upright or torso-dominant activities may suffer due to motion artifact and interference from surrounding musculature. We compared ECG signal quality during treadmill walking, circuit training, and an obstacle course using three chest-worn commercial devices (Polar H10, Equivital EQ-02, and Zephyr BioHarness 3.0) and a multi-lead ECG monitor (BIOPAC). Signal quality was quantified using the beat signal quality index (SQI), and HR data were rejected if SQI fell below 0.7 or if values were physiologically implausible. Signal rejection rate was calculated as the proportion of low-quality observations across device and activity type. Significant effects of both device (*p* < 0.001) and activity (*p* < 0.001) were observed, with greater signal rejection during the obstacle course and circuit training compared to treadmill walking (*p* < 0.01). The Zephyr exhibited significantly higher rejection rates than the Polar (*p* = 0.018) and BIOPAC (*p* = 0.017), while the Polar showed lower average rejection rates across all activities. These findings underscore the importance of including dynamic, non-upright tasks in ECG validation protocols and suggest that certain commercial devices may be more robust under realistic conditions.

## 1. Introduction

Multi-lead electrocardiogram (ECG) systems are the gold standard for measuring RR intervals [1], but due to their cost, complexity, and discomfort, commercial wearable ECG devices are more commonly used outside the laboratory setting. Many commercial wearable ECG monitors, such as Polar H10, Equivital EQ-02, and Zephyr Bioharness, are widely used in research to measure heart rate (HR) and heart rate variability (HRV) derived from RR intervals, across a range of physical activities [2,3,4]. Polar H10, Equivital EQ-02, and Zephyr Bioharness use similar ECG electrode sensing technology and have extended battery life, lightweight form, and operational ruggedness [5,6]. While the validity of these devices has been frequently evaluated [7,8,9], most validation studies have focused on upright activities (e.g., walking or running), leaving uncertainty about the effect of non-upright or torso-dominant movements on the quality of the ECG signal.

Recent research, however, has begun to examine ECG signal quality, the proportion of the cardiac signal unaffected by noise or motion artifacts, from chest-based monitors during non-upright activities, particularly those involving upper extremity movement or dynamic torso engagement. Although chest placement generally provides strong contact and good signal-to-noise ratios under resting or steady-state conditions, the integrity of ECG signals can deteriorate when users engage in dynamic or upper-body-dominant movements [10,11,12]. These may introduce motion artifacts that compromise signal quality, affecting the fidelity of derived metrics such as HRV and other beat-by-beat cardiac features.

Specifically, Širaiy et al. systematically evaluated ECG signal quality during maximal exercise testing and demonstrated that activity type substantially influences signal integrity [10]. During treadmill walking, ECG signal quality ranged from 58.9% to 62.2% (based on the proportion of clearly detectable QRS complexes as verified by both automated detection and medical visual inspection), whereas significantly higher quality (75.9% to 87.8%) was observed on a cycle ergometer, an activity with less torso movement [10]. These results demonstrate the disruptive effect of upper-body dynamics and emphasize that postural stability plays a critical role in maintaining reliable ECG signals.

Supporting this, Chaudhary et al. found that torso orientation and movement significantly affect ECG recordings, suggesting that anatomical changes during varied postures can alter both signal morphology and interpretability [13]. Additional work by Apandi et al. further highlighted the challenge of motion artifacts, noting that ECG signals collected during exercise are frequently compromised by unpredictable movement-induced noise, particularly during high-intensity or non-ambulatory activities [14]. Collectively, these studies emphasize the need for the assessment of signal quality in chest-worn ECG monitors across diverse activity types, in order to gauge fidelity during upper-body or non-upright movements.

Typically, multi-lead ECG devices have been used as the criterion device against which commercial wearable device accuracy is measured [15,16]. However, previous work comparing ECG signal quality from a multi-lead ECG vs. a leading commercial wearable device (Polar H10) during maximal exercise found that the commercial device provided higher-quality RR interval data, though it relied on visual analysis of the signal and did not assess HR or HRV [12]. Examining the effect of non-upright or torso-dominant movements on signal quality across commonly used commercial monitors such as the Polar H10, Equivital EQ-02, Zephyr BioHarness 3.0, vs. a multi-lead ECG device, could also provide researchers with evidence-based guidance for selecting the most appropriate device for robust ECG monitoring during real-world activities.

Therefore, the purpose of this report is to (1) characterize differences in ECG signal quality across increasingly complex, high-intensity activities via quantification of signal rejection rates (proportion of ECG data that is discarded due to not meeting predefined quality thresholds), and (2) to evaluate how ECG signal quality varies across different ECG devices including BIOPAC three-lead ECG and three widely used commercial monitors (Polar H10, Equivital EQ-02, and Zephyr BioHarness 3.0). Based on previous research showing improved signal quality during activities with minimal upper-body movement [10,12,17], we hypothesize that conditions involving non-upright postures and increased upper-body engagement will result in reduced signal quality.

## 2. Materials and Methods

The data presented in this report were collected as part of a previously published validation study of commercial and military-grade wearable ECG monitors, which report in detail participant demographics, device specifications, data acquisition protocols, and signal processing methods [18,19].

### 2.1. Participants

Thirty-nine healthy individuals (male/female = 27/12, age = 22.4 ± 3.9 years, height = 173.7 ± 8.1 cm, weight = 70.1 ± 11.9 kg) completed the test protocol wearing one of the three commercial monitors. Recruited participants were eligible if they were between the ages of 18 and 38, physically fit according to the American College of Sports Medicine guidelines [20], performed minimum levels of regular physical activity as defined by the World Health Organization [21], and had no current lower extremity injuries, gait disorders, or tattoos around their wrists or forearms. The University of Maryland Institutional Review Board approved the protocol (IRB# 186513), and all participants provided written informed consent prior to participation.

### 2.2. Experimental Protocol

Participants were assigned using a randomized block design to one of three commercial ECG monitor groups: Polar H10 (Polar Electro, Kempele, Finland; *n* = 12), Equivital EQ-02 (Hidalgo Ltd., Cambridge, UK; *n* = 13), or Zephyr BioHarness 3.0 (Zephyr Technology Corporation, Annapolis, MD, USA; *n* = 14). While each participant wore one of three commercial ECG monitors, all participants (*n* = 39) also wore the three-lead BIOPAC ECG system (BN-EL50 electrodes and BN-RSPEC-T transmitter, Biopac Systems, Goleta, CA, USA) on the chest. All commercial devices were donned as outlined by manufacturers guidelines, while the electrodes of three-lead ECG were placed using a modified Einthoven’s triangle. Given our aim to compare device-level signal rejection across varied activities, the full BIOPAC dataset was retained for analysis to maximize statistical power and provide a more stable estimate of signal quality under different activity types. Participants were asked to refrain from strenuous exercise for 24 h as well as from consuming caffeine, alcohol, and tobacco 12 h before participation.

Participants then performed a series of physical activities starting with a ramped treadmill exercise, followed by a circuit of common bodyweight training exercises, and ending with an obstacle course based on the ‘Load Effects Assessment Program-Army’ (LEAP-A) course.

The treadmill protocol involved participants walking at a constant speed (3.3 mph for males and 3.0 mph for females) and raising the incline by 3% and 2.5% for males and females, respectively, every three minutes. Participants walked on the treadmill at increasing inclines until they reached 80% of age-predicted maximum HR calculated using the Fox equation (220−age). The Fox equation was used as it has been found to be the most accurate to measured maximum HR compared to other prediction equations [22]. When participants reached their target HR, incline was reduced to 0% and speed to 2.5 mph for 3 additional minutes.

Following 3 min of standing rest on the treadmill, participants began the functional exercises. A circuit of squats, step-ups, lunges, and push-ups was performed twice with each exercise lasting one minute and the participant immediately transitioning into the following exercise. After 8 min of functional exercises, the participant had at least 3 min of rest before beginning the obstacle course.

Participants then immediately began the obstacle course which involved climbing over 32-inch mats, performing 2 types of military crawls (20-inch low crawl and 26 inch-high crawl for 20 feet each), an agility hurdle task involving short sprints over 6-inch hurdles, and two weighted tasks: pulling a 90-pound sled backwards for 40 feet and jogging the same distance while carrying a 25-pound sandbag at waist height. Exercises were performed consecutively, with participants completing the full course twice, separated by a brief self-paced rest. A 3 min seated recovery followed the second trial while data collection continued.

### 2.3. Calculation of Signal Quality

Time-synchronization, signal filtering, and R-R interval computation were conducted using validated procedures described in our prior work [19], including GPS-based clock alignment, peak-to-peak interval extraction via MATLAB’s PhysioNet Toolbox (version R2024b, Mathworks, Natick, MA, USA), and removal of low-frequency baseline wander artifact using a fourth-order Butterworth high-pass filter with a cut-off frequency of 5 Hz. Signal quality was assessed using the Beat Signal Quality Index (bsqi) function from PhysioNet toolbox [23], which computes a window-based signal quality index (SQI) by comparing the agreement between two independent R-peak detection methods. In this analysis, R-peaks identified by the findpeaks function and the jqrs algorithm [24] were used as two R-peaks detection methods, which were input into the bsqi() to generate SQI values. The SQI was computed using a rolling window of 10 s with a 1 s increment. Data were then classified as ‘rejected’ if they met any of the following criteria in each rolling window: an SQI value below 0.7 [25], mean heart rate below 30 bpm or above 240 bpm, or a mean heart rate difference in successive windows exceeding 10 bpm.

The threshold of 0.7 for SQI was selected based on prior studies demonstrating that SQI values around 0.7 effectively distinguish clean from noisy ECG signals [26,27,28]. To ensure that a threshold of 0.7 was an adequate choice for the current work, we conducted supplemental validation using a subset of our dataset, specifically from the BIOPAC treadmill recordings. A threshold of 0.7 maximized agreement with visual assessments and minimized both false positives and false negatives in identifying low-quality segments. While this value may not represent the optimal cutoff for all device and activity combinations, it was applied uniformly in this study to enable a standardized comparative assessment of signal quality. Signal rejection rate was calculated as the proportion of rejected windows based on these criteria, relative to the total number of rolling windows for each device and activity type.

### 2.4. Statistical Analysis

All analyses were conducted in R (version 4.3.2; R Core Team, 2023). Participant demographics were assessed for normality using the Shapiro–Wilk test and for variance homogeneity using Levene’s test. As several variables violated assumptions of normality and homogeneity of variance, a non-parametric Kruskal–Wallis H test was used to evaluate differences across wearable groups.

Signal quality results are summarized and reported as mean ± standard error of the mean (SEM) across different device and activity types. Levene’s test confirmed significant heterogeneity of variances across Device × Task combinations; therefore, a non-parametric aligned rank transform (ART) ANOVA was used to evaluate the main effects and interaction between Device and Task on signal rejection rate. This method allowed for full factorial analysis using rank-transformed data while accommodating heterogeneity of variances.

Post hoc pairwise comparisons (including mean difference and 95% confidence intervals) were conducted using the Games-Howell test, which does not assume equal variances or equal sample sizes and includes correction for multiple comparisons. For each pairwise, comparison, Hedges’ G was calculated as a standardized measure of effect size. Statistical significance was defined as *p* < 0.05.

## 3. Results

Participant demographics are displayed in Table 1. No statistically significant differences were found for demographic variables across devices. Mean (±SEM signal rejection rates across devices and activities are presented in Figure 1. Aligned rank transform (ART) ANOVA revealed significant main effects of both activity (F(2,255) = 42.05, *p* < 0.001) and device (F(3,255) = 6.53, *p* < 0.001) on signal rejection rate. However, there was no significant interaction between activity and device (F(6,255) = 0.54, *p* = 0.78), indicating that the effects of each factor were independent, with activity type having a substantially stronger effect.

Games-Howell post hoc comparisons showed that treadmill walking produced significantly lower signal rejection rates compared to both the exercise circuit (*p* < 0.001; g = −1.10) and the obstacle course (*p* < 0.0001; g = −1.32), indicating large effects. The obstacle course also had significantly higher rejection than the exercise circuit (*p* = 0.003; g = −0.42) reflecting a moderate effect. Across devices, the Zephyr BioHarness 3.0 had significantly higher rejection rates than both the BIOPAC (*p* = 0.017; g = −0.59) and the Polar H10 (*p* = 0.018; g = −0.70), each representing large effects. Differences between the remaining device pairs were not statistically significant, see Table 2.

## 4. Discussion

In this brief report, we quantified ECG signal quality by calculating the signal rejection rate across a range of activity types in three commercial monitors and a multi-lead ECG system. We used signal rejection rate, computed as the proportion of rolling windows with SQI values below 0.7, as an indirect measure of ECG quality by capturing the frequency of segments with inconsistent R-peak detection. Devices with lower signal rejection rates were interpreted as providing higher-quality, more stable signals suitable for downstream analysis, whereas higher rejection rates indicated greater susceptibility to artifact and signal degradation.

On average, all devices demonstrated significantly higher signal rejection during both the circuit training (29.9%) and obstacle course (40.4%) compared to treadmill walking (11.4%), with the obstacle course also producing significantly higher rejection rate compared to circuit training. These findings are consistent with previous work demonstrating that torso dynamics and upper-body motion impair ECG signal quality during exercise [10,11,12,13,14,17]. Significant reduction in ECG signal quality increases the likelihood of incorrect or missed R-peak detections which are the basis for HR and HRV estimation. Clifford et al. demonstrated that an SQI < 0.7 is indicative of unreliable ECG segments which should be considered compromised for HR and HRV estimation [29]. Unlike the majority of prior studies which have investigated wearable ECG quality and/or accuracy in stationary or upright walking activities [30,31,32], our inclusion of complex, non-upright activities such as crawling, pushups, sled pulls etc., revealed that over 30% of observations had SQI values below 0.7 across all devices during high-motion activities like circuit training and obstacle course racing. These results bring into question the potential robustness of ECG monitors to record with high fidelity during non-laboratory based activities, as while a device may still report HR during poor signal conditions, low signal quality reduces confidence in any derived metrics.

Our results, however, did demonstrate that some devices showed consistently greater signal quality across all activities. The 3-lead BIOPAC and Polar H10 devices had significantly lower signal rejection rates compared to the Zephyr Bioharness suggesting greater robustness to factors that can induce signal noise. Of note, there was no difference in signal rejection rate between the BIOPAC and Polar H10 device, with the Polar device demonstrating slightly lower rejection rates across activity types. This is consistent with findings by Gilgen-Ammann et al., who found that the Polar H10 had similar signal quality (as defined by the proportion of correctly detected RR intervals) relative to a 3-lead Holter ECG during low intensity activities like sitting and walking, but greater signal quality during high-intensity activities like jogging and strength training [12]. While multi-lead ECG devices are widely regarded as the gold-standard device for HR and HRV measurement in clinical and laboratory settings, these findings suggest that the Polar H10 may be a viable alternative for field-based applications, particularly during complex, high intensity activities that involve significant upper-body motion.

Various factors may drive loss of signal quality in non-upright or torso-dependent activities. These tasks often involve direct mechanical interference at the chest-strap interface, changes in electrode contact pressure, and substantial upper-body muscle activation, all of which can introduce motion artifacts or transient electrode detachment [33,34]. Teferra et al. demonstrated that the size of textile electrodes significantly affects signal quality during various physical activities, emphasizing that larger electrodes may provide improved signal fidelity across different movements [35]. Similarly, Wang et al. highlighted that the effectiveness of electrode materials and designs under physical motion influences observed ECG signal quality, underlining material science as a critical factor in the development of ECG technology [36]. Additionally, electrode positioning and contact integrity are critically important as studies have shown that the effectiveness of ECG monitors can be susceptible to placements that fluctuate with user movements [37]. This may partially explain the lower signal rejection rate of the Polar H10 device to the other commercial devices as its strap-based form factor with electrodes on the anterior chest/sternum may maintain more stable contact with the skin compared to the harness-based form factors of the Equivital EQ-02 and Zephyr Bioharness 3.

In addition to electrode design, placement, and material properties, signal processing methods play a critical role in mitigating motion-related signal degradation. Recent work has demonstrated that advanced algorithms, such as those incorporating wavelet transforms, can effectively isolate ECG features from overlapping noise sources like electromyographic (EMG) activity. These approaches have shown promise in enhancing signal quality during high-motion tasks, where conventional filtering techniques may be insufficient [34]. Integrating such signal processing strategies may therefore complement hardware improvements, providing a dual pathway to improving ECG data reliability in dynamic, field-relevant conditions. Although the signal processing algorithms used in the three commercial devices studied here are proprietary, differences in these algorithms (as well as sampling rates) may contribute to the observed variability in signal rejection rates across devices.

This report has several limitations. Each participant wore only one commercial ECG device, limiting direct within-subject comparisons. While we did not examine the precise motion-induced artifacts or movement patterns responsible for reduced SQI, the consistently elevated signal rejection rates observed during circuit training and obstacle course tasks across all devices provide clear, quantitative evidence that these activity types present greater challenges to ECG signal integrity that could impact derived metrics like HRV [38]. Additionally, we acknowledge that the SQI threshold of 0.7 was validated using BIOPAC data during the treadmill activity only. Although this value was chosen based on prior literature and our own analysis (balancing agreement with visual inspection while minimizing false positives and negatives) it may not generalize across devices or activity types. While the use of a fixed, detector-agnostic threshold promotes methodological consistency and is supported by prior work [26,28], it may obscure device-or task-specific differences in ECG signal quality. Future studies should explore device- and activity-specific tuning as more annotated data become available. Lastly, while the exercise circuit and obstacle course included a mix of upright and non-upright tasks, variability in movement type within each condition limits interpretation of task-specific effects. More pronounced differences in signal rejection may emerge when directly comparing discrete activities (e.g., walking vs. push-ups). Future work should investigate signal quality across specific task types to better determine the feasibility of using wearable ECG systems in upper-body dominant or non-upright operational environments.

## 5. Conclusions

These findings demonstrate the effect of non-upright, torso-intensive, high intensity activities on ECG signal quality. The large signal rejection rates seen across all devices in the exercise circuit and obstacle course suggest that HR and HRV estimation may be compromised during these types of activities, particularly within short analysis windows where noise or artifact may transiently obscure accurate R-peak detection. Therefore, researchers and clinicians must carefully evaluate ECG signal quality when using wearable devices to measure HR in high-motion conditions, as relying solely on reported HR values without signal verification may lead to misleading conclusions.

Additionally, it highlights the importance of including non-upright, torso-intensive activities in the validation of new ECG devices, as protocols limited to treadmill walking may fail to capture meaningful signal quality degradation under real-world conditions. Novel devices intended for operational use should be tested in scenarios that replicate the full range of body positions and movements encountered in applied settings. Furthermore, comparative validation against devices with demonstrated robustness, such as the Polar H10, which demonstrated greater signal quality across activity types in our study, may serve as a more practical benchmark than traditional lab-based systems, particularly in environments where reference-grade equipment is impractical or unreliable.

## Figures and Tables

**Figure 1 sensors-25-05186-f001:**
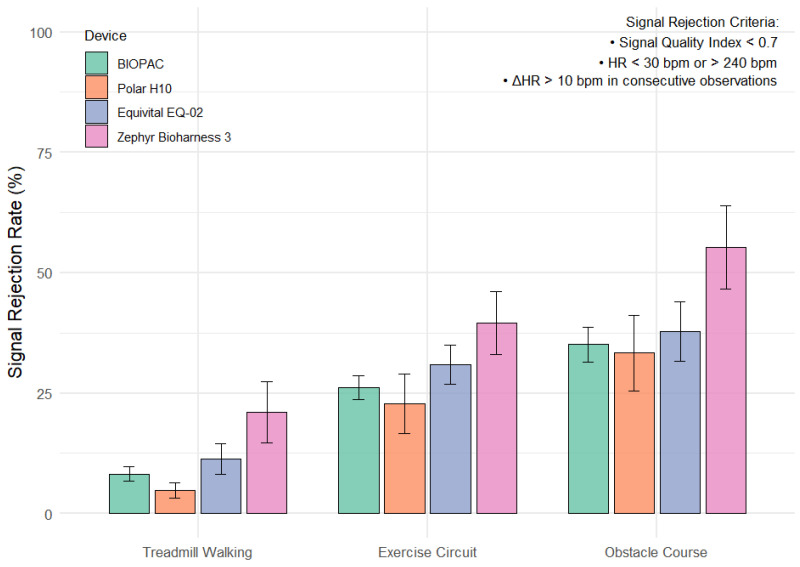
Mean signal rejection rate (%) by device across activity type. Signal rejection rate was defined as the proportion of heart rate data excluded due to low signal quality index (SQI < 0.7), physiologically implausible values, or abrupt beat-to-beat heart rate changes. Bars represent mean ± standard error.

**Table 1 sensors-25-05186-t001:** Participant demographics across device group. All participants wore the BIOPAC device and another randomly selected commercial wearable ECG device selected via block randomization with a between-subjects factor.

Device	*n*	Male/Female	Age (Years)	Height (cm)	Mass (kg)	BF%
BIOPAC	39	27/12	22.4 ± 3.9	173.7 ± 8.1	70.1 ± 11.9	17.6 ± 7.9
Polar	12	8/4	22.4 ± 2.6	173.0 ± 8.2	67.3 ± 10.6	17.2 ± 6.6
Equivital	13	10/3	24.2 ± 5.7	175.4 ± 6.6	71.8 ± 10.6	17.8 ± 9.1
Zephyr	14	10/4	20.6 ± 1.2	172.2 ± 9.5	70.9 ± 14.3	17.8 ± 9.4

**Table 2 sensors-25-05186-t002:** Pairwise comparisons of signal rejection rate by device and task using Games-Howell post hoc tests. * indicates statistically significant difference.

Factor	Comparison	Mean Difference	95% CI	Effect Size (g)	*p* Adj
Device	BIOPAC—Polar H10	−2.86	−14.2 to 8.5	0.13	0.908
	BIOPAC—Equivital EQ-02	3.54	−6.07 to 13.1	−0.17	0.766
	BIOPAC—Zephyr Bioharness 3.0	15.50	2.17 to 28.80	−0.59	0.017 *
	Polar H10—Equivital EQ-02	6.39	−19.60 to 6.83	−0.30	0.582
	Polar H10—Zephyr Bioharness 3.0	18.30	2.33 to 34.30	−0.70	0.018 *
	Equivital EQ-02—Zephyr Bioharness 3.0	11.9	−2.98 to 26.8	−0.47	0.161
Task	Treadmill Walking—Exercise Circuit	18.3	12.5 to 24.2	−1.1	<0.001*
	Treadmill Walking—Obstacle Course	28.1	20.6 to 35.6	−1.32	<0.001 *
	Exercise Circuit—Obstacle Course	9.79	1.54 to 18.0	−0.42	0.003 *

## Data Availability

The data will be shared upon reasonable request after all the necessary approvals (including IRB and sponsor) are obtained.

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
