# Peer review of "Activity Type Effects Signal Quality in Electrocardiogram Devices"

_sensors, 2025, doi:10.3390/s25165186_

Round 1

Reviewer 1 Report

Comments and Suggestions for Authors

The authors’ proposal is really interesting, however, it is not clear what the contribution of the research is to the State Of The Art. The authors state that the data presented was collected as part of it own previous published validation study. Time-syncronization, signal filtering, and R-R interval computation, including GPS-based clock alignment, high-pass filtering, etc., were conducted using validated procedures describe in his previous work too. It seems that the authors present just an extension of the previously reported work. 

Author Response

Reviewer 1

The authors’ proposal is really interesting, however, it is not clear what the contribution

of the research is to the State of The Art. The authors state that the data presented was

collected as part of it own previous published validation study. Time-synchronization, signal

filtering, and R-R interval computation, including GPS-based clock alignment, high-pass

filtering, etc., were conducted using validated procedures describe in his previous work

too. It seems that the authors present just an extension of the previously reported work.

Highlights

In the highlights section, authors should include simple text that allows the reader to

quickly and easily find information about the manuscript. Two bullet points are required;

not the description of the highlights or the questions posed. I recommend reviewing the

highlights section.

Keywords

I think that "wearable ECG devices" rather than only "wearable" will be more suitable

keyword. Specially considering that the manuscript is about ECG devices. The same

occur with the keyword "Signal Quality" were "ECG Signal Quality" will be more suitable.

Response: Thank you for your review and constructive feedback. We appreciate the time you took to evaluate our manuscript. Regarding your comment on the manuscript's contribution to the state of the art: While it is true that the dataset was collected as part of a previously published validation study, we respectfully clarify that this manuscript represents a distinct and novel analysis. We have emphasized this distinction more clearly in the revised manuscript to highlight that the goal of this report was not to reassess device accuracy but to quantify signal rejection rates using objective signal quality index (SQI) thresholds across a range of real-world tasks. Additionally, this work is submitted as a Brief Report, and deliberate consideration was given to using this format to succinctly present a focused, applied analysis that informs future validation and deployment of wearable ECG technologies in dynamic environments.

Regarding the Highlights section, thank you for this suggestion. We have revised this section (line 17) to include only two bullet points using simplified, informative language in accordance with the journal’s formatting guidelines. We also removed the instructional description from the template, which had been inadvertently left in.

For the Keywords, we appreciate the specific recommendations. We have revised the keywords (line 39) to use "wearable ECG devices" and "ECG signal quality" to more accurately reflect the manuscript content and improve discoverability.

Reviewer 2 Report

Comments and Suggestions for Authors

General comment:

This study examined the accuracy of several commercially available electrocardiogram (ECG) devices. A distinctive feature of this study is that verification was conducted during exercise rather than at rest.

The reviewers fully agree with the authors' assertion that ECG verification protocols should include upper body exercise.

Specific comments:

Comment 1:

In this study, ECG data is rejected if it meets any of the following three conditions:

  • The SQI value below 0.7
  • Mean heart rate below 30 bpm or above 240 bpm
  • Mean heart rate difference in successive windows exceeding 10 bpm.

The rejection rate for each condition were shown in Fig 1. The reviewer considered that the reasons for rejection (the above three criteria) should also be indicated. Please consider this.

Comment 2:

The results of this study indicate that the quality of ECG signals output from heart rate sensors is low, especially during intense exercise. This differs from the accuracy of the heart rate (or R-R interval) output of these sensors. To avoid any misunderstanding, the author should emphasize this distinction.

Comment 3:

I don't think the three criteria for rejection are overly strict. However, even the Polar H10, which performed best in this study, showed a rejection rate of 20–30% during circuit training and obstacle courses. This is not considered to be of sufficient quality. Therefore, in the conclusion, the authors should emphasize more strongly that researchers must pay sufficient attention to ECG quality when measuring heart rate (HR) during intense upper-body movement. Please consider this.

Author Response

The rejection rate for each condition were shown in Fig 1. The reviewer considered that the reasons for rejection (the above three criteria) should also be indicated. Please consider this.

Response: Thank you for the suggestion. We have added our rejection threshold to our figure as we believe that adds clarity to the figure.

The results of this study indicate that the quality of ECG signals output from heart rate sensors is low, especially during intense exercise. This differs from the accuracy of the heart rate (or R-R interval) output of these sensors. To avoid any misunderstanding, the author should emphasize this distinction.

Response: This is an excellent suggestion as this is an important distinction. We have added language in the introduction to clarify (line 98) as well as in the discussion (line 213).

I don't think the three criteria for rejection are overly strict. However, even the Polar H10, which performed best in this study, showed a rejection rate of 20–30% during circuit training and obstacle courses. This is not considered to be of sufficient quality. Therefore, in the conclusion, the authors should emphasize more strongly that researchers must pay sufficient attention to ECG quality when measuring heart rate (HR) during intense upper-body movement. Please consider this.

Response: We have added language at the end of our conclusion to emphasize this, as we also believe this is an important note for clinicians.

Reviewer 3 Report

Comments and Suggestions for Authors

This manuscript, "Activity Type Effects Signal Quality in Electrocardiogram Devices" contains several significant issues that require careful revision.

Topic of evaluating performance of wearable ECG devices under various physical activities is not particularly novel. Numerous studies have already assessed ECG signal quality during movement and impact of motion artifacts across different types of wearable monitors. Though inclusion of "non-upright" activities such as obstacle courses and circuit training is somewhat less explored, this alone does not sufficiently justify this study (brief report) as addressing a significant gap.

What it adds to field is unclear, as many prior works have already compared commercial ECG devices in real-world or simulated environments. Without more in-depth exploration of specific motion-induced artifacts or unique methodological advancement, manuscript lacks innovative edge required for this brief report publication.

Manuscript provides minimal detail on experimental setup. Number of participants, characteristics of study population, duration of data collection, and criteria for data exclusion (beyond SQI threshold) are not specified. These omissions make it difficult to assess the generalizability and reproducibility of results.

Use of single signal quality threshold cutoff of 0.7 across all devices and activities appears arbitrary. There's no indication that this threshold was validated or compared against ground truth signals, nor is there any explanation for physiological plausibility criteria. This weakens confidence in accuracy of the rejection rate metric.

Although p-values are reported, there is no clear indication of corrections for multiple comparisons (e.g., Bonferroni), which is essential when performing multiple pairwise tests. The reliability of statistical significance is therefore questionable.

Inclusion of three chest-worn commercial devices and one clinical monitor (BI-OPAC) is presented without explaining why these specific models were chosen, or whether they represent different technological classes. This limits interpretability of comparative analysis.

Conclusion appears overstated given limited data and simplistic analytical framework. For example: the statement that Polar device “demonstrated the most consistent performance across tasks” is not supported by robust evidence. No measure of consistency (e.g., intra-device variability, signal stability across repetitions or users) is reported.

Claim that findings “underscore importance of including dynamic, non-upright tasks in ECG validation protocols” is valid in principle, but the data presented do not delve deeply enough into mechanics of signal degradation under such movements to substantiate the claim.

In short, conclusions are only loosely tied to presented evidence, and key questions—such as why Polar performed better or how motion artifacts differ across activities—remain unaddressed.

Manuscript is dense and mechanical in tone. Terms like “signal rejection rate” are used without defining baseline expectations or implications. This suggests that authors prioritized reporting over communication.

No clinical or practical implications are discussed in this manuscript therefore misses an opportunity to translate findings into real-world guidance—for example, for clinicians monitoring patients during rehabilitation, or athletes using wearable tech.

Comments on the Quality of English Language

The English could be improved to more clearly express the research.

Author Response

Topic of evaluating performance of wearable ECG devices under various physical activities is not particularly novel. Numerous studies have already assessed ECG signal quality during movement and impact of motion artifacts across different types of wearable monitors. Though inclusion of "non-upright" activities such as obstacle courses and circuit training is somewhat less explored, this alone does not sufficiently justify this study (brief report) as addressing a significant gap.

Response: We agree that the broader topic of wearable ECG performance and motion artifact has been explored in the literature. However, our study specifically addresses an under-studied gap in this body of work, the quantitative impact of upper-body-intensive, non-upright activities on signal quality across multiple commercial ECG devices. Such activities may be performed by firefighters who are increasingly utilizing wearables to monitor cardiac activity on the fireground as part of efforts to mitigate adverse events. While prior studies have noted that motion can degrade ECG signal quality, most have focused on upright, steady-state activities such as walking, running, or cycling, scenarios in which electrode contact and torso stability are relatively well-maintained. What distinguishes our brief report is not merely the inclusion of non-upright tasks like obstacle courses and circuit training, but the systematic quantification of signal degradation using a standardized signal quality index (SQI) and statistical comparisons across both device and activity. We also tested multiple torso-worn ECG monitors, each with distinct form factors and proprietary algorithms, alongside a reference-grade multi-lead system. This design enhances the practical relevance of our findings and offers actionable guidance for researchers, clinicians, and developers. Additionally, we believe it is useful to inform future development, and validation attempts to account for non-resting or upright activities.

What it adds to field is unclear, as many prior works have already compared commercial ECG devices in real-world or simulated environments. Without more in-depth exploration of specific motion-induced artifacts or unique methodological advancement, manuscript lacks innovative edge required for this brief report publication.

Response: Thank you for this additional comment. We appreciate your perspective and agree that many studies have explored the performance of commercial ECG devices in various settings. However, our brief report is intentionally scoped to provide a focused, applied contribution: specifically, to quantify and compare the extent of signal degradation, via signal rejection rates, across common ECG devices during realistic, full-body activities that are underrepresented in the literature. Rather than offering new signal processing techniques or artifact characterization algorithms, our contribution lies in the application of a standardized SQI-based rejection metric across both device and activity type. This offers a practical framework for benchmarking device robustness that is directly relevant to researchers and practitioners selecting ECG tools for use outside controlled lab environments. Additionally, we acknowledge the potential value in deeper exploration of specific motion-induced artifacts. However, such an analysis would require a different study design and scope (e.g., high-resolution synchronized kinematics and ECG waveform inspection) and would be more appropriate for a full-length methodological paper. The findings presented here were part of a larger validation study, and we felt it was important to leverage existing data from this funded project to share a focused, yet meaningful, analysis that adds value to the field.

Manuscript provides minimal detail on experimental setup. Number of participants, characteristics of study population, duration of data collection, and criteria for data exclusion (beyond SQI threshold) are not specified. These omissions make it difficult to assess the generalizability and reproducibility of results.

Response: Thank you for this feedback. We respectfully note that the number of participants and key sample characteristics are provided in the manuscript (table 1), and that the full study protocol, including participant demographics, experimental procedures, device specifications, and data exclusion criteria beyond the SQI threshold, is thoroughly described in our previously published validation study (Lindsey et al., 2025, BMJ Mil Health), which we cite on line 105. Given the brief report format, we intentionally referenced this larger validation study to avoid redundancy and maintain focus on the novel signal quality analysis presented here. However, in response to your comment, we have added a few additional details in the Methods section (including a table of participant demographics by group) to better support transparency and reproducibility without exceeding the intended scope of this format (line 169). If more details are desired, we would be happy to provide in a future revision

Use of single signal quality threshold cutoff of 0.7 across all devices and activities appears arbitrary. There's no indication that this threshold was validated or compared against ground truth signals, nor is there any explanation for physiological plausibility criteria. This weakens confidence in accuracy of the rejection rate metric.

Response: Thank you for raising this point. We did validate this value, but did not include that in the manuscript which was an oversight. We have included this information in 180 and 288.

Although p-values are reported, there is no clear indication of corrections for multiple comparisons (e.g., Bonferroni), which is essential when performing multiple pairwise tests. The reliability of statistical significance is therefore questionable.

Response: We thank the reviewer for this important point. While we did not use Bonferroni correction explicitly, we performed post hoc comparisons using the Games-Howell test, which is a robust method that adjusts for multiple comparisons and does not assume equal variances or sample sizes. This approach is consistent with our use of Welch’s ANOVA for the main effects. We have also added language on line 202 to specifically call this out.

Inclusion of three chest-worn commercial devices and one clinical monitor (BI-OPAC) is presented without explaining why these specific models were chosen, or whether they represent different technological classes. This limits interpretability of comparative analysis.

Response: Thank you for this helpful feedback. We selected three chest-worn commercial devices because chest-worn sensors have been shown to be more accurate than wrist-based devices, particularly during high-intensity activities (Wang et al., 2017). Although the form factors differ slightly, Polar (single strap), Zephyr (single-shoulder harness), and Equivital (dual-shoulder harness), all three devices are chest-worn and use ECG electrode technology. This positions them within the same general technological class, in contrast to wrist- or finger-worn devices that typically use photoplethysmography (PPG) (Hinde et al., 2021). These specific devices were selected based on their suitability for high-intensity, outdoor activity monitoring. Our selection criteria prioritized extended battery life, lightweight design, and operational ruggedness to ensure usability in demanding environments. To clarify that these devices are from a similar technological class and to justify the selection of these devices we added a sentence to the introduction in lines 50-52. As stated in the first line of the introduction, multi-lead electrocardiogram systems are the gold standard. To clarify this when we first introduce the Biopac in line 97, we added text to state it is typically the gold standard.

Wang R, Blackburn G, Desai M, et al. Accuracy of wrist- worn heart rate monitors. JAMA Cardiol 2017;2:104–6.

Hinde K, White G, Armstrong N. Wearable devices suitable for monitoring twenty four hour heart rate variability in military populations. Sensors (Basel) 2021;21

Conclusion appears overstated given limited data and simplistic analytical framework. For example: the statement that Polar device “demonstrated the most consistent performance across tasks” is not supported by robust evidence. No measure of consistency (e.g., intra-device variability, signal stability across repetitions or users) is reported.

Response: We appreciate the reviewer’s thoughtful feedback. We agree that the original phrasing may have overstated the findings. Our analysis was based on average signal rejection rates across activity types and did not include formal measures of intra-device variability or signal stability across repetitions or participants. We have therefore revised the language in the conclusion to more accurately reflect the scope and limitations of our findings. We specifically changed our language on line 34 and 311 to reflect this.

Claim that findings “underscore importance of including dynamic, non-upright tasks in ECG validation protocols” is valid in principle, but the data presented do not delve deeply enough into mechanics of signal degradation under such movements to substantiate the claim.

Response: Thank you for this observation. We agree that a mechanistic analysis of how specific movements contribute to ECG signal degradation would strengthen the physiological interpretation of these findings. However, our intent was not to isolate artifact sources at the movement level, but rather to demonstrate that signal quality is consistently and significantly compromised during upper-body and non-upright activities, relative to standard treadmill walking. The phrase “underscore the importance of including dynamic, non-upright tasks in ECG validation protocols” is grounded in our empirical observation that signal rejection rates increased markedly during such tasks across all devices. While we do not dissect the biomechanical or signal-level origins of degradation in this brief report, our statistical comparisons by activity type offer clear evidence that these activities pose challenges to signal quality, and that they should therefore be represented in validation testing. This issue has been added as a limitation to inform future research studies.

In short, conclusions are only loosely tied to presented evidence, and key questions—such as why Polar performed better or how motion artifacts differ across activities—remain unaddressed.

Response: Thank you for this comment. We acknowledge that our conclusions focus on observed trends rather than mechanistic explanations. As noted in prior responses, the scope of this brief report was intentionally limited to quantifying differences in signal rejection rates across devices and activities, not to fully investigate the underlying causes of those differences. Future studies using synchronized motion tracking and waveform inspection will be needed to explore the biomechanical and device-specific factors responsible for these effects.

Manuscript is dense and mechanical in tone. Terms like “signal rejection rate” are used without defining baseline expectations or implications. This suggests that authors prioritized reporting over communication.

Response: Thank you for this feedback. When responding to previous reviewer feedback, we added to the “Calculation of Signal Quality” section a description of the signal quality rejection rate that is typically used to determine clean vs. noisy EMG signals. We added to line 61 of the introduction this definition when we first mention the concept of signal-to-noise (still need to add in if you think that makes sense-I left a comment in manuscript at this area). We also tried to remove technical jargon such as “QRS” in the introduction to make the manuscript less dense and mechanical and clearly define terms when first introduced to improve readability and flow.

No clinical or practical implications are discussed in this manuscript therefore misses an opportunity to translate findings into real-world guidance—for example, for clinicians monitoring patients during rehabilitation, or athletes using wearable tech.

Response: Thank you for this suggestion. We added to the conclusion (lines 314-319) that researchers and clinicians must consider signal quality when measuring heart rate during higher intensity activities.

Round 2

Reviewer 3 Report

Comments and Suggestions for Authors

The authors have attempted to address the suggestions and comments; however, the overall improvement remains limited.  

Comments on the Quality of English Language

The English could be improved to more clearly express the research.

Author Response

Addition of references to a prior study for methodological details is noted. However, for sake of reproducibility and transparency, essential elements such as participant characteristics, device placement, and session duration should be briefly summarized within current manuscript itself—not only cited. Readers of a brief report may not consult prior publication, and thus standalone clarity and replicability remain insufficient.

Response: We appreciate the reviewer’s emphasis on reproducibility and transparency. In our initial revision, we added details on participant characteristics (see Table 1). We have also clarified that commercial devices were donned according to manufacturer recommendations and specified electrode placement for the 3-lead ECG (lines 135–137). With respect to session duration, we have intentionally not included this information because it is not critical to interpreting or replicating the present study’s findings and is described in detail in the cited publication. While we understand the reviewer’s concern that some readers may not consult the cited paper, we are of the opinion that readers seeking full procedural detail will reference the original source, as is common scholarly practice.

 Revised manuscript includes mention of validating 0.7 SQI threshold, but without detailed explanation or comparative analysis with ground truth electrocardiogram signals. It remains unclear whether this threshold is universally appropriate across different devices and activities. Providing at least supplemental figure or brief explanation of threshold’s empirical basis would strengthen confidence in rejection rate metric.

Response: We appreciate the reviewer’s concern regarding the generalizability of the 0.7 SQI threshold. To clarify, the threshold was chosen based on prior literature demonstrating its ability to differentiate clean from noisy ECG segments, and we further validated it against visually scored BIOPAC treadmill data from our own dataset. While we recognize that signal characteristics can vary by device and activity type, we intentionally applied a uniform 0.7 threshold across all conditions to enable consistent, “out-of-the-box” comparison of signal quality between devices (lines 188-192).

Use of Games-Howell post hoc comparisons is appropriate; however, manuscript should explicitly state how Type I error was controlled across multiple device and activity comparisons. Even with robust tests, manuscript lacks any discussion of effect sizes or confidence intervals, which are essential for interpreting practical significance of statistical findings—especially when claiming device superiority.

Response: Thank you for the feedback. Our original manuscript included CIs for post-hoc comparisons but we have also included effect sizes now. Also, to explore the potential interaction effect between device and activity we implemented an aligned rank transform ANOVA.

Justification for selecting specific chest-worn devices is more clearly articulated in revision, which is appreciated. However, further detail about their technical specifications—such as sampling rate, electrode type, and signal processing algorithms—would improve the interpretability of between-device differences.

Response: We appreciate the reviewer’s suggestion to include additional technical specifications for the chest-worn devices. As this work is presented in a brief report format, our aim was to focus on the primary study objective—examining the effect of activity type on ECG signal quality—while keeping the manuscript concise. Including full details on sampling rates, electrode types, and proprietary signal processing algorithms would require substantial expansion beyond the scope and format of this report. For readers interested in these technical specifications, they are readily available in the manufacturers’ documentation and prior validation studies, which we have cited.

Despite changes, conclusion still tends to overgeneralize findings. For example, stating that one device “performed best” requires a clearer definition of what constitutes “performance”—and whether such results hold across repetitions, users, and different environmental conditions. Without intra-device consistency data or waveform-level validation, such claims remain tentative.

Response: We appreciate the reviewer’s concern regarding potential overgeneralization in our conclusions. In revising the manuscript, we have replaced the phrase “performed best” with more precise language indicating that one or more devices demonstrated lower signal rejection rates, and therefore greater signal quality, compared to others. This terminology directly reflects our operational definition of “performance” in the present study—i.e., proportion of usable ECG data—and avoids implying broader claims about device accuracy or applicability beyond our testing conditions. We acknowledge that our results do not include intra-device consistency data or waveform-level validation, and we have been careful to frame our findings within these methodological boundaries. While the reviewer suggests that without such additional data the conclusions must remain tentative, we believe the revised language accurately represents the scope of our study and provides clear context for interpreting the observed differences.

Added lines discussing need for clinicians and researchers to consider signal quality during dynamic tasks are mentioned. However, manuscript still underplays the practical implications of findings. How might these results influence choice of electrocardiogram monitors in rehabilitation, firefighting, or athletic training contexts? What are risks of relying on lower-quality signals in such settings? This would enhance real-world applicability which is totally missing in the revised version of this manuscript.

Response: Thank you for the feedback. We have added significant  language in both the discussion and text to outline the practical findings. This can be found at lines 264 – 277, 287 – 291, and 352 – 358.

Tone remains overly technical, and terms like “signal rejection rate” are used without adequate contextualization. Consider including a brief explanation of what constitutes an “acceptable” or “poor” rejection rate in practical terms. In addition, discussing how much signal loss is tolerable for meaningful clinical or research use would help readers interpret findings.

Response: Thank you for the feedback. We have fully defined signal rejection rate in section 2.3 and have then used it as well as signal quality throughout the manuscript to describe our findings. The reviewer’s point on defining what constitutes poor or acceptable rejection rate is well received. We included literature that investigated at what threshold of SQI does an ECG signal become unreliable for HR and HRV estimation (lines 264 – 267). However, we believe that defining ‘good’ or ‘acceptable’ levels or signal rejection rate will depend on the application of the ECG monitoring, i.e., real-time cardiac abnormalities or summary exercise reports, which to define is outside the scope of this manuscript.

Manuscript emphasizes differences in rejection rates but does not explore why these differences occur. A brief discussion of likely mechanisms (e.g., electrode displacement, muscle artifacts, harness design) would add explanatory value. While deeper mechanistic study may be outside scope of this report, acknowledging potential contributors to variability can strengthen the interpretation of results.

Response: Thank you for the feedback. We added much detail in our revision on the potential sources for differences in signal quality (lines 300-326).